

# *Pdk3*'s role in RANKL-induced osteoclast differentiation: insights from a bone marrow macrophage model

Nan Zhang[1], Lingting Wang[2] and Xuxin Ye[3]

[1] College of Physical Education, Anhui Normal University, Wuhu, China
[2] Spinal Surgery, The First Affiliated Hospital of Wannan Medical Collage, Wuhu, China
[3] Office of Hospital Admission and Discharge, The First Affiliated Hospital of Wannan Medical Collage, Wuhu, China

Corresponding author
Nan Zhang, 1582313000@qq.com

## ABSTRACT

**Background.** Osteoporosis (OP) is a chronic disease characterized by decreased bone mass, loss of skeletal structural integrity and increased susceptibility to fracture. Available studies have shown that the pyruvate dehydrogenase kinase (PDK) family is associated with osteoclastogenesis and bone loss, but the specific role of *Pdk3* in bone pathology has not been systematically investigated.

**Methods.** A cell OP model was established in receptor activator for nuclear factor-$\kappa$B Ligand (RANKL)-induced bone marrow macrophages (BMMs). Hereafter, the expression levels of *Pdk3* and osteoclastogenesis feature genes including nuclear factor of activated T cells 1 (*Nfatc1*), Cathepsin K (*Ctsk*), osteoclast associated Ig-like receptor (*Oscar*) in BMMs-derived osteoclasts were examined based on real-time quantitative PCR and western blotting methods. Further, the phosphorylation of ERK, P65 and JAK/STAT and their correlation was *Pdk3* was gauged. In particular, changes in the activity of these signaling pathways were observed by silencing experiments of the *Pdk3* gene (using small interfering RNA). Finally, the effects of *Pdk3* gene silencing on signaling pathway activity, osteoclastogenesis, and related inflammatory and apoptotic indicators were observed by transfection with PDK3-specific siRNA.

**Results.** Following RANKL exposure, the levels of *Pdk3* and osteoclastogenesis feature genes were all elevated, and a positive correlation between *Pdk3* and osteoclastogenesis feature genes was seen. Meanwhile, ERK, P65 and JAK/STAT phosphorylation was increased by RANKL, and *Pdk3* was confirmed to be positively correlated with the phosphorylation of ERK, P65 and JAK/STAT. Additionally, in RANKL-exposed osteoclasts, *Pdk3* knockdown diminished the phosphorylation of ERK, P65 and JAK/STAT, reduced the expressions of osteoclastogenesis feature genes. Importantly, knockdown of *Pdk3* also reduced the expression of inflammatory cytokines and resulted in elevated levels of *Bax* and *Casp3* expression, as well as downregulation of *Bcl2* expression.

**Conclusion.** This study reveals for the first time the role of Pdk3 in RANKL-induced osteoclastogenesis and OP. These findings provide a foundation for future studies on the role of Pdk3 in other bone diseases and provide new ideas for the development of OP therapeutics targeting Pdk3.

## INTRODUCTION

Osteoporosis (OP) refers to a chronic condition which is featured by the decrease in bone mass, the loss of skeletal integrity and the increase of susceptibility to fracture, which, according to some relevant data, affects 10.2% of adults over 50 years old and is expected to increase to 13.6% by the year 2030 (*Ramchand & Leder, 2024*; *Harris, Zagar & Lawrence, 2023*). Currently, pharmacotherapy has been recommended as the primary therapeutic option for patients who have suffered from the fragility fracture, including stimulants of bone matrix formation, inhibitors of bone resorption and dual-action drugs (*Iolascon et al., 2020*; *Khosla & Hofbauer, 2017*; *Marie, 2006*). However, despite the effectiveness in the therapy of OP, the early diagnosis of OP remains a challenge and a concern (*Yalaev et al., 2022*). Currently, the main tools used clinically for the treatment of OP include synovitis ointment, bisphosphonates, hormone replacement therapy, and biologics, and although these therapies are effective in reducing bone loss, they are often accompanied by side effects and have limited long-term efficacy (*Muñoz, Robinson & Shibli-Rahhal, 2020*; *Yu & Wang, 2022*; *Zhang et al., 2023*). Molecular insights on the pathogenesis of OP, therefore, are required so as to work out the clinically viable therapy regimens.

The principal reason accounting for the development of processes underlying the progression of OP has been categorized to the imbalance between the functions of osteoblasts and osteoclasts (*Kaur, Nagpal & Singh, 2020*). Osteoblasts are those remarkably versatile cells building up our skeleton which require tight regulation in all phases of their differentiation in order to ensure proper skeletal development and homeostasis (*Ponzetti & Rucci, 2021*). However, osteoclasts are derived from the monocyte/macrophage lineage and are responsible for the resorption of aging bone. These cells fuse to form multinucleated giant cells with bone-resorbing capabilities (*Chen et al., 2018*; *Da, Tao & Zhu, 2021*). The bone formation and resorption is in a stable at physiological conditions; however, the triggering of bone metabolic diseases can lead to an imbalance of abnormal bone structure or function (*Zaidi, 2007*). Currently, evidence suggests that a direct interaction between osteoblasts and osteoclasts facilitates the bidirectional transmission of activation signals *via* EFNB2-EPHB4, FASL-FAS, or SEMA3A-NRP1, which plays a vital role in regulating the differentiation and survival of both cell types. Additionally, osteoblasts secrete various molecules such as M-CSF, RANKL/OPG, WNT5A, and WNT16, which can either encourage or inhibit the differentiation and maturation of osteoclasts (*Kim et al., 2020*; *Tonna et al., 2014*; *Wang et al., 2015*). Hence, these findings prompt us to explore the underlying mechanisms by which osteoclasts exert their effects in osteoporosis or other bone diseases. Pyruvate dehydrogenase kinases (PDKs, four genes: *PDK1-PDK4*) are the major regulatory enzymes of glucose metabolism due to their negative role in the regulation of pyruvate dehydrogenase complex (PDC) *via* phosphorylation (*Anwar et al., 2021*). While linking PDKs with bone, some existing studies have suggested that *PDK4* induction leads to bone loss *via* promoting osteoclastogenesis (*Wang et al., 2012*). In the meantime, *Pdk1* was shown to be required for the function of bone marrow hematopoietic stem and progenitor cells in transplantable mice and to be able to influence osteoclast differentiation in ankylosing spondylitis (*Halvarsson, Eliasson & Jönsson, 2017*;

*Sun et al., 2021*). Additionally, some researchers have demonstrated that the prevention of osteoporosis in mice lacking estrogen is achieved through the inhibition of *Pdk2*, which likely works by diminishing irregular activation of osteoclasts, potentially through the suppression of the nuclear factor-$\kappa$B ligand (RANKL)-CREB-cFOS-nuclear factor of activated T cells 1 (*Nfatc1*) signaling pathway (*Lee et al., 2021*). Notably, unlike other members of the PDK family, the specific role of Pdk3 in bone pathology has not been systematically investigated.

Here, this study preliminarily initiates with the aim to delve into the specific involvement of *Pdk3* in osteoclasts. We explored the regulation of osteoclastogenic signature gene expression, signaling pathway activation, inflammatory response and apoptosis by *Pdk3* through gene silencing experiments, thus initially revealing the potential mechanism of *Pdk3* in OP. Our study not only provides new insights into the specific role of *Pdk3* in osteoclast function but also lays the theoretical foundation for the future development of *Pdk3*-targeted therapeutic strategies for OP.

## MATERIALS AND METHODS

### Bone marrow macrophages

Bone marrow macrophages (BMMs) were purchased from Shanghai Fuheng Biotechnology Co., LTD. (Shanghai, China). Hereafter, the non-adherent cells were layered onto a Ficoll density gradient solution and centrifuged at 440 g for 30 min at ambient temperature. The cells were cultured in $\alpha$-minimum essential medium (12000063, Gibco, Waltham, MA, USA) with 10% bovine calf serum (F2442, Sigma, Burlington, MA, USA) and 1% penicillin-streptomycin (P4333, Sigma) at 37 °C under 5% $CO_2$.

Cultured BMMs were inoculated in 6-well plates with $5\times 10^5$ cells per well. Macrophage colony-stimulating factor (M-CSF, M9170, Sigma) at 30 ng/mL was added to the culture medium to promote cell survival and proliferation. Subsequently, 50 ng/mL of RANKL (R0525, Sigma) was added 48 h after the initial culture and the culture was continued for 5 days to induce the differentiation of BMMs to osteoblasts. This is mainly based on the ability of RANKL to activate osteoclast differentiation and activity by binding to the RANK receptor, leading to an increase in bone resorption and thus triggering OP (*Xiao et al., 2015*).

The small interfering RNAs (siRNAs) specific to *Pdk3* (hereafter si-Pdk3#1 and si-Pdk3#2) as well as the corresponding negative control were synthesized by Guangzhou RiboBio Co., Ltd (Guangzhou, China). Next, the transfection of cells was performed with the Lipofectamine 2000 reagent (11,668; Thermo Fisher, Waltham, MA, USA) following the guidelines provided by the manufacturer. The relevant target sequence was displayed in Table 1.

### Western blot

The total protein in our cultured BMMs were isolated using a commercial RIPA lysis buffer (R0010, Solarbio, Beijing, China), followed by the quantification of the concentration. Protein concentration was determined using the BCA Protein Quantification Kit (Pierce, Appleton, WI, USA) to ensure a consistent amount of protein was loaded into each sample.
**Table 1  Target sequences for transfection.**

| Target | Target sequence (5′–3′) |
| --- | --- |
| si-NC | AGAGGAAATAATAATCATGAAGG |
| si-Pdk3#1 | AAGGGATAATGCATGTGAAAAAA |
| si-Pdk3#2 | AGGGATAATGCATGTGAAAAAAC |

Equal amounts of protein were separated by SDS-PAGE, transferred to a polyvinylidene fluoride membrane (YA1701, Solarbio) (Millipore, Bredford, USA) and probed with primary antibodies against phosphorylated-ERK1/2 (1:2000, CST), ERK1/2 (1:10000, Abcam), phosphorylated-P65 (1:1000, CST), P65 (1:1000, Abcam), phosphorylated-JAK1 (1:10000, Abcam), JAK1 (1:10000, Abcam), phosphorylated-STAT1 (1:10000, Abcam), STAT1 (1:10000, Abcam) and housekeeping control GAPDH (1:10000). Then the membranes were further incubated with a solution containing horseradish peroxidase (1:5,000, GE Healthcare, Chicago, IL, USA)-labeled secondary antibody at ambient temperature for 2 h and exposed to the electrochemical luminescence reagent (PE0010, Solarbio) to develop protein bands. ImageJ 1.42 quantified the density of protein bands.

## Total RNA extraction and real-time quantitative PCR

The total RNA was isolated using the TriZol assay kit (15596026, Invitrogen, Waltham, MA, USA) and then reverse transcribed into cDNA using the PrimeScript RT Reagent Kit (RR037Q, Takara, Shiga, Japan). Then the PCR was conducted using the QuantiTect SYBR Green RT-PCR Kit (204243, Qiagen, Hilden, Germany) at the following conditions: 95 °C for 15 s, and 35–45 cycles of 94 °C for 15 s, 60 °C for 30 s and 72 °C for 30 s. The experiment was carried out in triplicate. The primer sequences used are shown in Table 2. The relative gene expression was calculated by the $2^{-\Delta\Delta Ct}$ method with GAPDH as the housekeeping control (*Livak & Schmittgen, 2001*).

## Statistical analysis

SPSS 21.0 (SPSS, Inc., Armonk, NY, USA) software was applied in statistical analysis. The data of three independent trials were expressed as mean ± standard deviation. The student's *t*-test was applied for two-group comparison throughout the study, and the Pearson's correlation test was applied in correlation analyses. In this study, statistical significance was set at $P < 0.05$.

## RESULTS

### Involvement of *Pdk3* in RANKL-induced osteoclastogenesis *in vitro*

RANKL has been shown to be a key regulator of osteoclast differentiation, survival and activity. RANKL attaches to its receptor, RANK, to trigger the primary signaling pathways responsible for osteoclast formation, thereby promoting both transcriptional and epigenetic processes essential for osteoclastogenesis (*Park-Min, 2018*; *Yasui et al., 2011*; *Bae et al., 2023*). To this end, we used an *in-vitro* OP model was constructed using the RANKL as the inducer in BMMs, and the expression levels of *Pdk3* as well as osteoclastogenesis feature genes were the gauged. A sharp increase in *Pdk3* level was clearly seen following

**Table 2  Primer sequences for qPCR.**

| Target | Primer sequence (5′–3′) | |
| --- | --- | --- |
| | Forward | Reverse |
| Pdk3 | CTATCAAACAGTTCCTGGAC | CTTTAACCACATCAGCTACA |
| Nfatc1 | TACTTGGAGAATGAACCTCT | CAGTAAAAACCTCCTCTCAG |
| Trap | GCACAGATTGCATACTCTAA | GCTGGTCTTAAAGAGTGATT |
| Ctsk | AGACTCACCAGAAGCAGTAT | CTGGAGTAACGTATCCTTTC |
| Oscar | ATACTCCAGCTGTCGACTC | AGCAGTTCCAGAACATTACT |
| Bax | TGAACAGATCATGAAGACAG | TCTTGGATCCAGACAAGC |
| Bcl2 | CATTATAAGCTGTCACAGAGG | GGAGAAATCAAACAGAGGTC |
| Casp3 | AAGAACTTCCATAAGAGCAC | AGGTGCTGTAGAGTAAGCAT |
| Il1b | CTGAACTCAACTGTGAAATG | AAGTCAATTATGTCCTGACC |
| Il6 | GTCTTCTGGAGTACCATAGC | TATCTGTTAGGAGAGCATTG |
| Tnf | CTCACACTCAGATCATCTTCTC | TTCTCCTGGTATGAGATAGC |
| Gapdh | GCTTAGGTTCATCAGGTAAA | TGACAATCTTGAGTGAGTTG |

RANKL exposure (Fig. 1A, $P < 0.01$), concurrent with the elevation of osteoclastogenesis feature genes *Nfatc1* (Fig. 1B, $P < 0.01$), *Trap* (Fig. 1C, $P < 0.001$), *Cathepsin K* (*Ctsk*, Fig. 1D, $P < 0.001$), *osteoclast associated Ig-like receptor* (*Oscar*, Fig. 1E, $P < 0.01$). We observed a positive correlation between *Pdk3* and osteoclastogenesis feature genes *Nfatc1* (Fig. 1F, $R^2 = 0.697$, $P = 0.039$), *Trap* (Fig. 1G, $R^2 = 0.917$, $P = 0.003$), *Ctsk* (Fig. 1H, $R^2 = 0.902$, $P = 0.004$), and *Oscar* (Fig. 1I, $R^2 = 0.688$, $P = 0.041$). These findings support a possible important regulatory role for PDK3 in osteoclastogenesis, which provides a basis for further investigation of its role in OP.

## Positive correlation between *Pdk3* and relevant signaling pathways in RANKL-induced osteoclastogenesis *in vitro*

Hereafter, the involvement of signaling pathways relevant to osteoclastogenesis like ERK, P65, and JAK/STAT was determined by western blot. Increased level of phosphorylated-ERK1/2 was seen following RANKL intervention (Figs. 2A–2B, $P < 0.01$), which was positively correlated with *Pdk3* (Fig. 2C, $R^2 = 0.695$, $P = 0.039$). Further, RANKL exposure also resulted in significant upregulation of phosphorylation of P65 (Figs. 2D–2E, $P < 0.01$), JAK1 (Figs. 2G–2H, $P < 0.01$) and STAT1 (Figs. 2J–2K, $P < 0.01$), which were all positively correlated with *Pdk3* (Figs. 2F, 2I, 2L).

## *Pdk3* knockdown led to the inactivation of relevant signaling pathways in RANKL-induced osteoclastogenesis *in vitro*

To explore the specific role of *Pdk3* in RANKL-induced osteoclastogenesis *in vitro*, we first verified that transfection of *Pdk3* in this cell was successful (Fig. 3A, $P < 0.001$).

The levels of proteins related to the signaling pathways were measured again with the presence of *Pdk3*-specific siRNA. Accordingly, the results of western blot have manifested the phosphorylation levels of all these proteins were significantly down-regulated compared to the control, including ERK1/2 (Figs. 4A–4B, $P < 0.001$), P65 (Figs. 4C–4D, $P < 0.001$), JAK1 (Figs. 4E–4F, $P < 0.01$) and STAT1 (Figs. 4G–4H, $P < 0.001$).

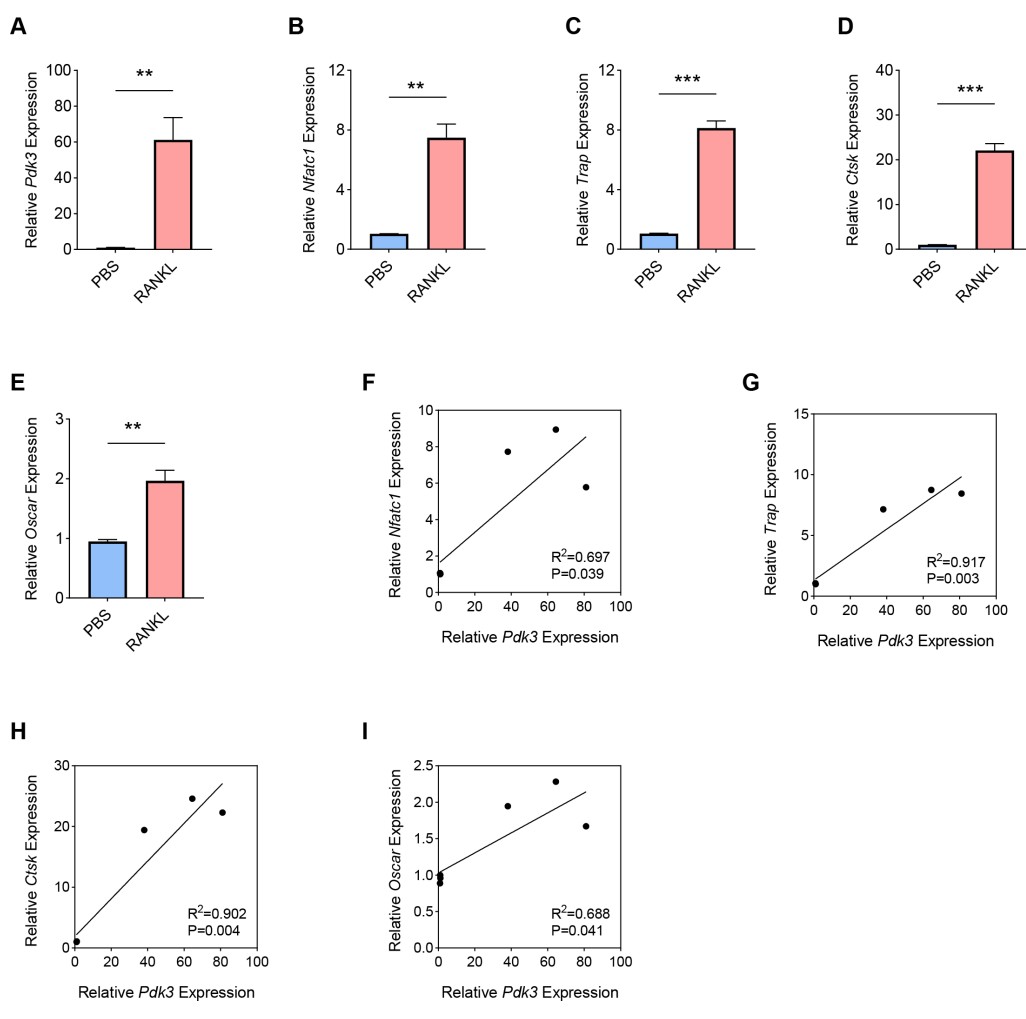

**Figure 1** **Involvement of *Pdk3* in RANKL-induced osteoclastogenesis *in vitro*.** (A) Expression level of *Pdk3* in RANKL-induced osteoclastogenesis *in vitro*. (B–E) Expression levels of osteoclastogenesis feature genes *Nfatc1* (B), *Trap* (C), *Ctsk* (D), and *Oscar* (E). (F–I) Correlation between *Pdk3* and osteoclastogenesis feature genes *Nfatc1* (F), *Trap* (G), *Ctsk* (H), and *Oscar* (I). All experimental data of three independent trials were expressed as mean ±standard deviation. ** $P < 0.01$, *** $P < 0.001$.

## Effects of *Pdk3* silencing on RANKL-induced osteoclastogenesis *in vitro*

The expressions of osteoclastogenesis feature genes were then quantified to further examine the effects of *Pdk3* silencing on RANKL-induced osteoclastogenesis *in vitro*. According to the relevant results of qPCR, we observed that Pdk3 silencing significantly downregulated the levels of genes characteristic of osteoclastogenesis relative to controls, including *Nfatc1* (Fig. 5A, $P < 0.001$), *Trap* (Fig. 5B, $P < 0.001$), *Ctsk* (Fig. 5C, $P < 0.0001$), *Oscar* (Fig. 5D, $P < 0.0001$).

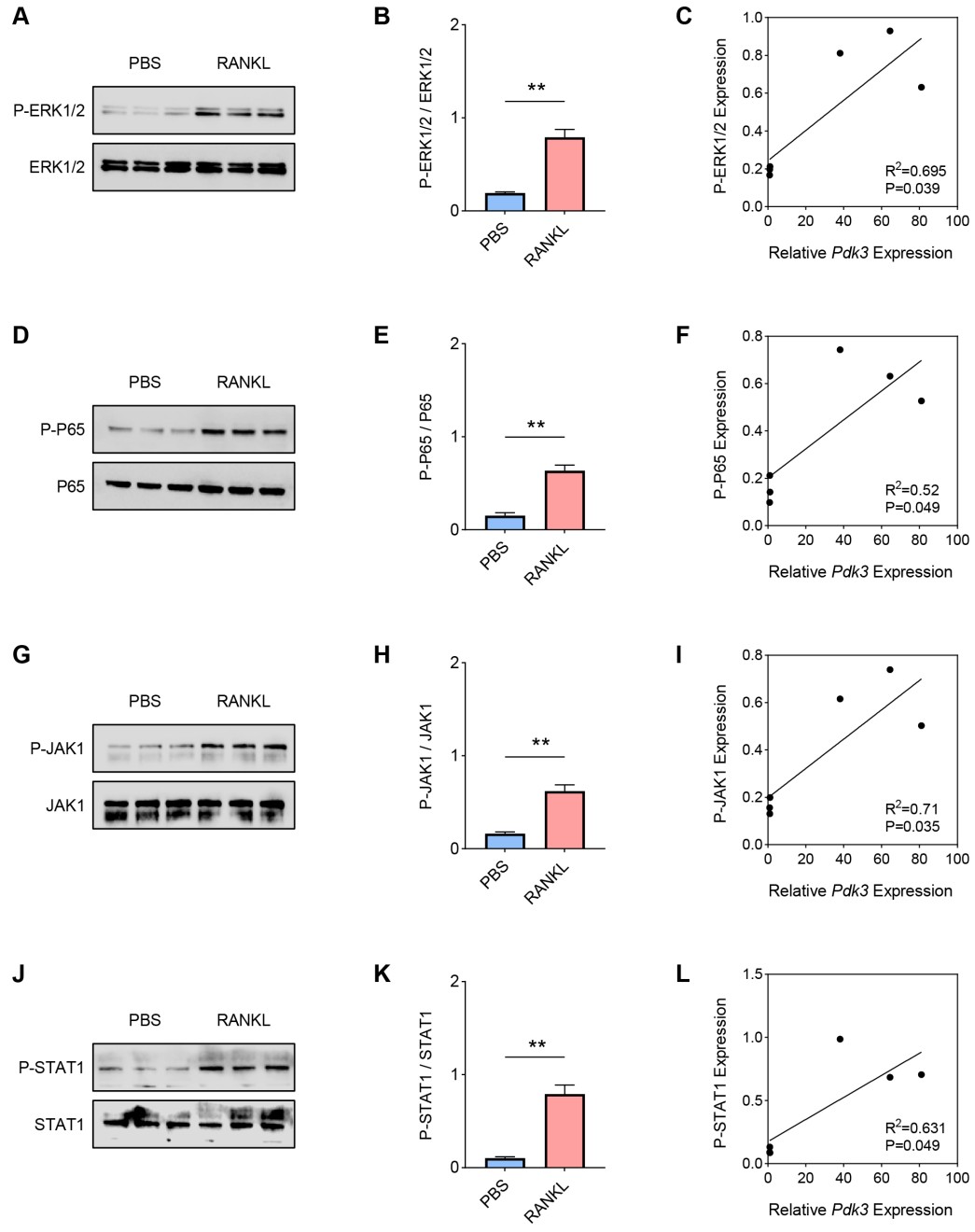

**Figure 2** **Positive correlation between *Pdk3* and relevant signaling pathways in RANKL-induced osteoclastogenesis *in vitro*.** (A–B) Phosphorylation level of ERK1/2 in RANKL-induced osteoclastogenesis *in vitro*. (C) Correlation between phosphorylation level of ERK1/2 and *Pdk3*. (D–E) Phosphorylation level of P65 in RANKL-induced osteoclastogenesis *in vitro*. (F) Correlation between phosphorylation level of P65 and *Pdk3*. (G–H) Phosphorylation level of JAK1 in RANKL-induced osteoclastogenesis *in vitro*. (I) Correlation between phosphorylation level of JAK1 and *Pdk3*. (J–K) Phosphorylation level of STAT1 in RANKL-induced osteoclastogenesis *in vitro*. (L) Correlation between phosphorylation level of STAT1 and *Pdk3*. All experimental data of three independent trials were expressed as mean ±standard deviation. **$P < 0.01$.

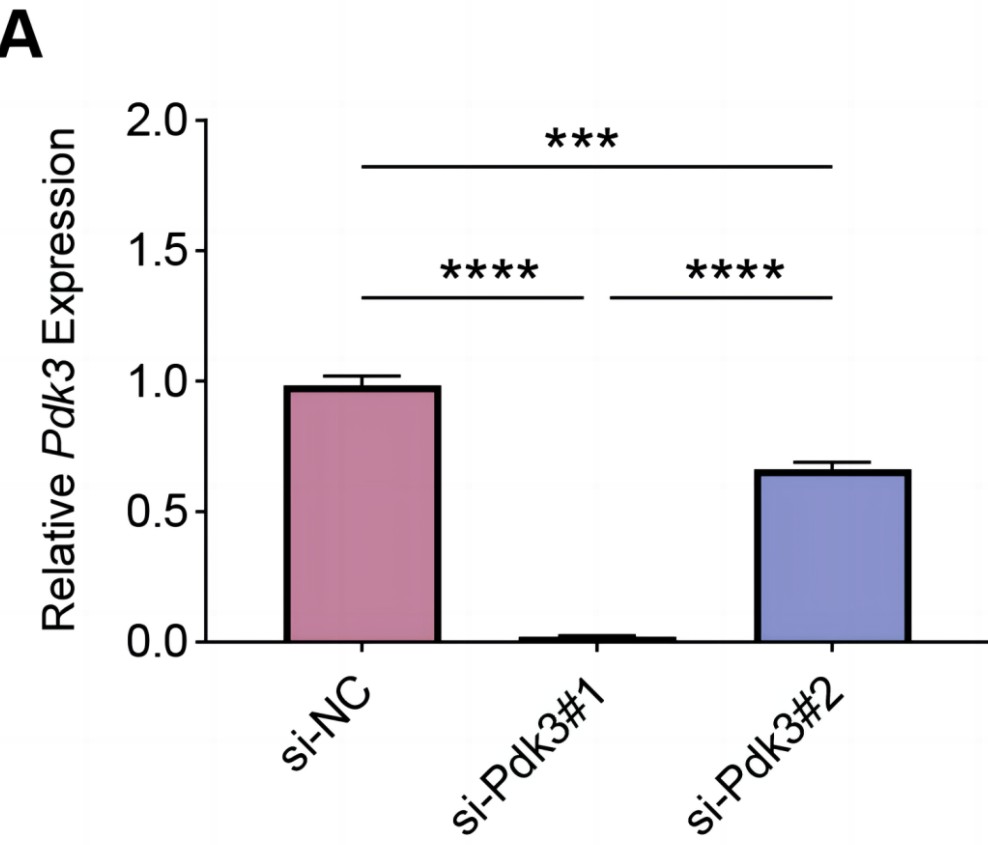

**Figure 3 Knockdown efficiency validation.** (A) *Pdk3*-specific small interfering RNAs were applied for the transfection and the knockdown efficiency was tested. All experimental data of three independent trials were expressed as mean ±standard deviation. ***$P < 0.001$, ****$P < 0.0001$.

### Effects of *Pdk3* silencing on RANKL-induced apoptosis and inflammation *in vitro*

Finally, we determined the effects of *Pdk3* silencing on RANKL-induced apoptosis and inflammation *in vitro*. It was clearly seen that *Pdk3* silencing led to the elevation of *Bax* and *Casp3* expression yet suppressed that of *Bcl2* (Figs. 6A–6C, $P < 0.01$). Relevant results on the inflammatory cytokines expressions have additionally proven that *Pdk3* silencing led to the suppression on all the inflammatory cytokines including *Il1b*, *Il6* and *Tnf* (Figs. 6D–6F, $P < 0.001$).

## DISCUSSION

The specific effects and mechanism of *Pdk3* regulation in osteoclastogenesis have not been systematically interpreted, which thus provides us an opportunity to commence this research to fill the blank. In our current study, we first found genetic evidence in RANKL-induced BMMs that *Pdk3* gene plays a crucial role in osteoclastogenesis. In other words, *Pdk3* expression was proven to be elevated in response to RANKL exposure, and

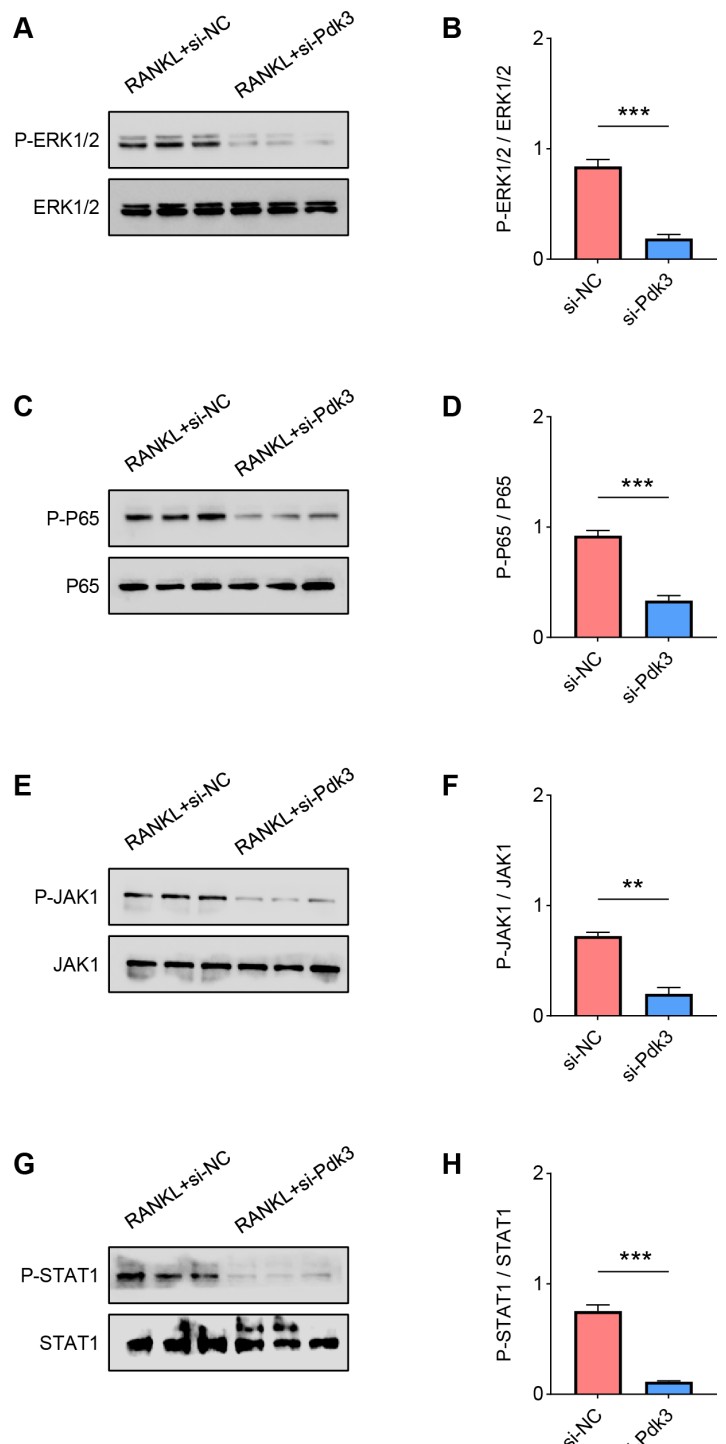

**Figure 4** *Pdk3* **knockdown led to the inactivation of relevant signaling pathways in RANKL-induced osteoclastogenesis *in vitro*.** (A–B) Following the silence of *Pdk3*, the quantified phosphorylation level of ERK1/2 in RANKL-induced osteoclastogenesis *in vitro*. 

**Figure 4 (…continued)**
(C–D) Phosphorylation level of P65 after the knockdown of *Pdk3* in RANKL-induced osteoclastogenesis *in vitro*. (E–F) After the transfection of *Pdk3*-specific siRNA, the level of JAK1 phosphorylation in RANKL-induced osteoclastogenesis *in vitro*. (G-H) Phosphorylation level of STAT1 in RANKL-induced osteoclastogenesis *in vitro* following the silencing of *Pdk3*. All experimental data of three independent trials were expressed as mean ±standard deviation. **$P < 0.01$, ***$P < 0.001$.

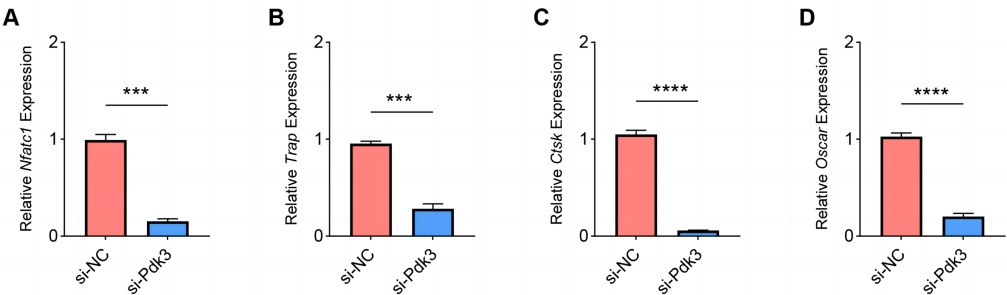

**Figure 5  Effects of *Pdk3* silencing on RANKL-induced osteoclastogenesis *in vitro*.** (A–D) Expression levels of osteoclastogenesis feature genes *Nfatc1* (A), *Trap* (B), *Ctsk* (C), *Oscar* (D) following the silencing of *Pdk3*. All experimental data of three independent trials were expressed as mean ±standard deviation. ***$P < 0.001$, ****$P < 0.0001$.

a positive correlation was seen in *Pdk3* and osteoclastogenesis feature genes *Nfatc1* (a master transcription factor required for osteoclast differentiation *Kang et al., 2020*), *Trap* (a gene critical to osteoclast activation *Takegahara, Kim & Choi, 2024*), *Ctsk* (a member of the papain family of cysteine proteases highly expressed by activated osteoclasts *Costa et al., 2011*), and *Oscar* (a regulator of osteoclast differentiation *Nedeva et al., 2021*). The subsequent assay results have confirmed that the silencing of *Pdk3* could repress the levels of these feature genes and attenuate the inflammation yet promote the apoptosis of BMMs-derived osteoclasts. Thus, our findings provide new insights into understanding the role of *Pdk3* in osteoclasts in OP and provide a theoretical basis for therapeutic strategies for OP patients.

Osteoclasts, multinucleated cells deriving from monocyte/macrophage-lineage cells and resorbing bone, have been documented to continuously destroy the bone in order to maintain the bone volume and calcium homeostasis throughout the lifespan of vertebrates (*Udagawa et al., 2021*). RANKL is the membrane-bound factor expressed by osteoclastogenesis-supporting cells like osteoblasts and osteocytes and critically involved in pathologic bone disorders (*Takayanagi, 2021*; *Sigl & Penninger, 2014*). Osteoclast precursors can express RANK (a known RANKL receptor), recognize RANKL expressed by the osteoblasts *via* cell–cell communication and differentiate into osteoclasts in the presence of M-CSF (*Udagawa et al., 2021*). RANKL/RANK pathway has been underlined to control osteoclasts activity and formation, which therefore has been identified as a key factor on bone turnover in diverse pathological conditions (*Amin et al., 2020*). Existing studies of studying osteoclast *in vitro* have extensively applied the technique of isolating osteoclast from primary BMMs or culturing the RAW264.7 cell lines, all

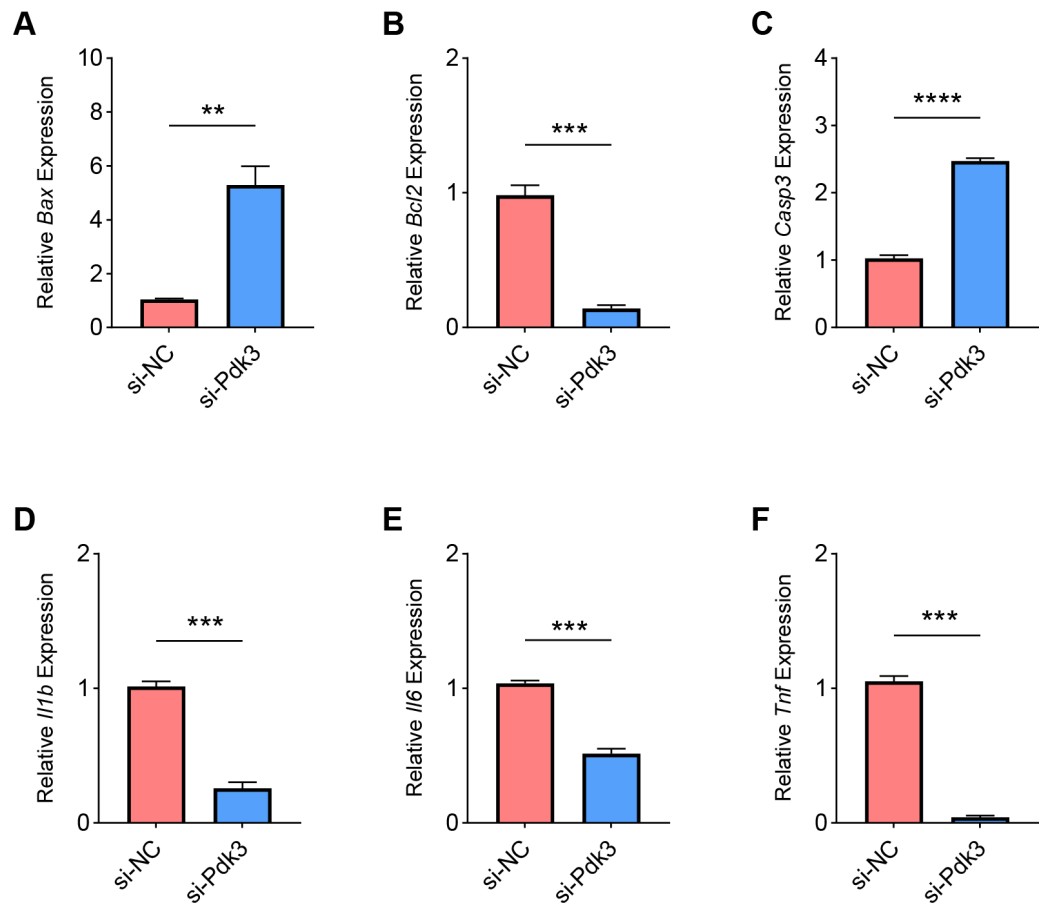

**Figure 6** **Effects of *Pdk3* silencing on RANKL-induced inflammation and apoptosis *in vitro*.** (A–C) Expression levels of apoptosis-related proteins *Bax* (A), *Bcl2* (B) and *Casp3* (C) in response of *Pdk3* knockdown. (D–F) Levels of inflammatory cytokines *Il-1 β* (D), *Il-6* (E) and *Tnf* (F) following the knockdown of *Pdk3*. All experimental data of three independent trials were expressed as mean ±standard deviation. **$P < 0.01$, ***$P < 0.001$, ****$P < 0.0001$.

of which have been widely engaged in bone homeostasis research (*Song et al., 2019*). The former technique of isolation was applied in our current study, and the isolated BMMs were treated with M-CSF and RANKL to induce an osteoclast-like cells so as to examine the effects of *Pdk3* on RANKL-induced osteoclastogenesis in OP *in vitro*. The study of *Lee et al. (2021)* has demonstrated that the deficiency of *PDK2*, another member of the PDK family, can prevent the ovariectomy-induced bone loss in mice *via* regulating the RANKL-NFATc1 pathway during osteoclastogenesis. Our current study, likewise, proved the involvement of *Pdk3* in RANKL-treated BMMs-derived osteoclasts. Specifically, following the confirmation that *Pdk3* was highly expressed in RANKL-treated BMMs-derived osteoclasts, the additional investigation has suggested that *Pdk3* knockdown could diminish the expression of osteoclastogenesis feature genes *in vitro*, which hinted the potential of *Pdk3* on osteoclastogenesis.

Osteoclastogenesis has been defined as an ongoing rigorous course including osteoclast precursors fusion and bone resorption executed by the degradative enzymes, which is also underscored to be controlled by and relevant to some processes like inflammation (*Tong et al., 2022*). Meanwhile, both survival and apoptosis are of major importance in the life cycle of osteoclasts, and the regulation of osteoclast apoptosis has been recognized as a critical factor in bone remodeling, where Bcl2 family member proteins and caspases have been shown to take part (*Soysa & Alles, 2019*; *Ke et al., 2019*; *Song et al., 2020*). In addition to the known endocrine, metabolic and mechanical factors, emerging evidences have further pointed out that inflammation also exerts significant influence on bone turnover, thus inducing OP (*Ginaldi, Di Benedetto & De Martinis, 2005*). Certain pro-inflammatory cytokines play possible critical roles both in normal bone remodeling process and in the pathogenesis of OP (*Ginaldi, Di Benedetto & De Martinis, 2005*). *IL-6*, for instance, promotes osteoclast differentiation and activation, while *IL-1* is another potent bone resorption stimulator linked to accelerated bone loss (*Manolagas, 2000*; *Wei et al., 2005*). Further, *TNF* is proven to play a pivotal role in osteoclast maturation (*Epsley et al., 2020*). Besides, *TNF* has been underscored to signal *via* NF-$\kappa$B and the MAPKs, and *Il6 via* the JAK-STAT pathway (*Osta, Benedetti & Miossec, 2014*). In RANKL-induced osteoblasts, the phosphorylation levels of JAK1 and STAT1 were significantly increased, suggesting that this signaling pathway was activated during osteoblast activation. Knockdown of the Pdk3 gene significantly inhibited the phosphorylation of JAK1 and STAT1, further suggesting that Pdk3 may affect the differentiation and activity of osteoblasts through the regulation of the JAK1-STAT1 signaling pathway. All these pathways have been revealed to be involved in OP, according to some relevant studies (*Li et al., 2022*; *Yang et al., 2023*; *Xu et al., 2018*). While trying to link the association between PDKs and these pathways, Tnf can promote the degradation of PDK4 in endothelial cells to support pro-inflammatory cytokines in a NF-$\kappa$B-dependent manner (*Boutagy et al., 2023*). In the meantime, another study on bladder cancer has stressed the anti-metastatic effects of *PDK4 via* the ERK and JNK pathways in bladder cancer cells (*Lee et al., 2022*). In our current study, we firstly proved the modulation of *Pdk3* on these signaling pathways in RANKL-induced BMMs-derived osteoclasts, as supported by the fact that *Pdk3* silencing diminished the phosphorylation of P65, ERK1/2 and JAK/STAT.

Nonetheless, it should be noted that there are some limitations to our study. First, all experiments in this study were performed in an *in vitro* model, and in the future by constructing *Pdk3* knockout or overexpression mouse models to be able to more validate its specific role in OP. In addition, we did not address other functional phenotypes of osteoclasts (*e.g.*, bone resorption capacity, cell migration and proliferation capacity, *etc*). Therefore, it is important to further expand the scope of the experiments to incorporate functional phenotyping experiments to be able to comprehensively assess the effects of *Pdk3* on osteoclast function. Finally, although *Pdk3*, as a member of the pyruvate dehydrogenase kinase family, may play an important role in cellular metabolism, its metabolic regulatory role in osteoclasts or osteoblasts was not explored in depth in this study. Future studies should explore the specific role of *Pdk3* in cellular metabolism by combining metabolic

analysis techniques, such as glycolytic flux assay and mitochondrial function assay. This will help to reveal the broader biological functions of *Pdk3* in osteoporosis.

So far as we are concerned, is the first to interpret the effect of *Pdk3* on OP *via* modulating the osteoclastogenesis using RANKL-induced BMMs. The relevant mechanisms of *Pdk3* were preliminarily explored to be related to the suppression of the phosphorylation of ERK, P65 and JAK/STAT, the reduced expressions of osteoclastogenesis feature genes, the attenuated inflammation-associated cytokines, and regulated the expression of apoptosis-related proteins. This study indeed opens up a novel avenue for OP prevention and provides a rationale for the development of therapies targeting *Pdk3*.

## Abbreviations

| | |
|---|---|
| **OP** | Osteoporosis |
| **PDKs** | Pyruvate dehydrogenase kinases |
| **PDC** | pyruvate dehydrogenase complex |
| **RANKL** | receptor activator for nuclear factor-$\kappa$B ligand |
| **Nfatc1** | nuclear factor of activated T cells 1 |
| **BMMs** | bone marrow macrophages |
| **M-CSF** | macrophage colony-stimulating factor |
| **siRNAs** | small interfering RNAs |
| **Ctsk** | Cathepsin K |
| **Oscar** | osteoclast associated Ig-like receptor |

### Funding

The authors received no funding for this work.

### Competing Interests

The authors declare there are no competing interests.

### Author Contributions

- Nan Zhang conceived and designed the experiments, analyzed the data, prepared figures and/or tables, authored or reviewed drafts of the article, and approved the final draft.
- Lingting Wang conceived and designed the experiments, performed the experiments, analyzed the data, prepared figures and/or tables, authored or reviewed drafts of the article, and approved the final draft.
- Xuxin Ye performed the experiments, analyzed the data, prepared figures and/or tables, authored or reviewed drafts of the article, and approved the final draft.

### Data Availability

The raw data is available at GitHub and Zenodo:

- Available at https://github.com/123Nan744/Raw-data-updated.git

- 123Nan744. (2024). 123Nan744/Raw-data-updated: Raw data updated (v.1.1.1). Zenodo. Available at https://doi.org/10.5281/zenodo.12661649.

## Supplemental Information

Supplemental information for this article can be found online at http://dx.doi.org/10.7717/peerj.18222#supplemental-information.

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
