# Peer review of "Pdk3’s role in RANKL-induced osteoclast differentiation: insights from a bone marrow macrophage model"

_PeerJ, doi:10.7717/peerj.18222_

## Round 0.1 · original submission · Major Revisions

After careful consideration of all the reviews, I have decided that a major revision is necessary before the manuscript can be considered for publication. The reviewers have raised several important points that need to be addressed. Please carefully consider all the reviewers' comments and suggestions in your revision. In your resubmission, please include a point-by-point response to each of the reviewers' comments, detailing how you have addressed their concerns or explaining why you disagree with any particular points.

Reviewer 1 ·

Basic reporting

The aim of this study was to investigate the molecular mechanisms affecting osteoporosis through a series of cellular experiments and molecular assays, especially from the phenotypic perspective of osteoclasts to reveal the relevant mechanisms. In this study, we first induced bone marrow macrophages with chemical reagents to establish an osteoporosis (OP) model, and determined and constructed the correlation between Pdk3 and genes related to osteoclast formation. After the subsequent construction of Pdk3-silenced cell lines, the phosphorylation levels and expression levels of ERK, P65 and JAK/STAT were clarified by molecular assays, laying the foundation for the revelation of the relevant molecular mechanisms of this gene-mediated osteoporosis. Subsequently, the expression of apoptosis and inflammation-related genes was evaluated by qPCR.
1. Since PDKs are major regulatory proteins of glucose metabolism, why did this study not set out to investigate the regulatory mechanisms of this protein in osteoclast glycolysis? There are many optional energy metabolism experiments that could be carried out, why did this study not take this direction? Please explain why and complete the introduction section.
2. The introduction section states that the clinical outcomes of OP are significant, so what is the significance of conducting this study? Can this study make a qualitative breakthrough in the clinical effectiveness of OP? Please state what is the specific intention of conducting this study.
3. What are the specific tools currently available for the clinical treatment of OP? What are the advantages and disadvantages of each of these tools? This study was not conducted in detail, and it is recommended that the introduction be supplemented to emphasize the distinctive intent of this topic.
4. The interaction between osteoblasts and osteoclasts is an important cause of osteoporosis, but the interaction between the two and the mechanism of the interaction are not described in detail in the introduction, and thus it is suggested that additional literature be added to this.

Experimental design

1. The study as a whole consisted of multiple molecular assay experiments, so why not perform molecular assays along with cellular experiments on the phenotypic revelation of osteoclasts? Is the cell proliferation and migration phenotype not important for osteoblasts? Please explain this.

Validity of the findings

1. The description in Figure 1 is too general and does not highlight what it means that Pdk3 levels are significantly elevated after RANKL exposure, as well as the elevation of osteoclast signature genes, and does not add a summary description at the end thus leaving the reader confused, and thus suggests that this should be supplemented and improved.
2. The effect of Pdk3 silencing on RANKL-induced osteoclastogenesis in vitro urgently needs to be explained, but the experimental results of the present study do not seem to reveal this argument in its entirety, e.g. do the osteoclast-associated genes measured in the present study encompass the full range of osteoclast characteristics? Please explain this in the results section and add analysis where necessary.

Additional comments

1. Does the research in this paper really fill the current gaps related to osteoclast research? Because the subjects of this paper were mined from the existing literature and there are no innovative results, it is therefore recommended that the opening part of the discussion be modified to make the summary of the results in this paper appear more modest.
2. The discussion of pdk3 in this paper is clearly insufficient, and it is recommended that the molecular assays in this paper be combined with an in-depth elaboration of the gene's function in osteoblasts, particularly highlighting whether the gene influences disease progression through phosphorylation levels, and glucose metabolism-related pathways.
3. The limitations of this paper are not only the lack of in vivo validation, but also the many shortcomings of the in vitro experiments, which are not described in this paper, which is a cop-out. Thus, it is recommended that the limitations of this paper be discussed in more detail, especially by stating the general idea of follow-up studies.

Reviewer 2 ·

Basic reporting

In this study, the author explores the potential link between the Pdk3 and the osteoclastogenesis in osteoporosis. Prognostic factors in osteoporosis were identified through bioinformatics methods. The experimental design is rigorous. However, there are still some deficiencies in details in the manuscript.
1. In the abstract, can the author simply explain the role of ERK, P65 and JAK/STAT pathways in osteoporosis.
2. Line 49, the specifically therapeutic drugs of osteoporosis should be supplemented.
3. What are the predisposing factors of osteoporosis, and what are the typical symptoms, diagnostic criteria of osteoporosis. Please add it in introduction.

Experimental design

no comment

Validity of the findings

no comment

Additional comments

4. What are several pathways and exact mechanism that affected the balance between the osteoblasts and osteoclasts.
5. Line 61, what are the specific bone metabolic diseases, what mechanisms do they promote the development of osteoporosis.
6. Line 81, The author please add the full name of RANKL and briefly explain the pathogenesis of osteoporosis induced by RANKL.
7. Line 146-148, the Pdk3 is a kinase, whether Pdk3 participates in phosphorylation of these pathways, and whether there are relevant reports
8. The role of JAK1-STAT1 affected the osteoclast in osteoporosis has not been discussed.
9. Osteoclasts are important factors in the occurrence of osteoporosis, so whether there are reports to achieve the intervention of osteoporosis by promoting the function of osteoblasts, what is the crosstalk between osteoblasts and osteoclasts.
10. Line 191-193, deficiency of PDK2 affected ovariectomy-induced bone loss, what is the verification process of this experiment? Can we determine whether PDK2 and PDK3 influence osteoclast formation in osteoporosis through the same mechanism.

Reviewer 3 ·

Basic reporting

Ambiguous in areas where content was presented but now scaffolded in the introduction nor referred to in the abstract.

Acceptable English but the literature references and background/context was not sufficient.
Suggest a background that more closely aligns with a RANKL induced in vitro model of osteoclast differentiation rather than the stretch to an osteoporosis model.
Content presented in the results was not introduced in the abstract or introduction.
Much of the required background could be moved from the discussion to better scaffold the study and clarify areas that are known or not yet known. It is stated how research fills the identified knowledge gap but this was not clearly linked to assessment outputs. If the introduction is refined and overstatements removed this would better align. As osteoporosis involves aging and hormonal mechanisms these are more osteoclasts or osteoblast dependent. Further, the disease involved increased resorption which was now assessed by these assays. The focus here is more mechanistic not osteoclast function of activity. This needs pointing out as a limitation when looking at the role of mechanistic factors.

Professional article structure
-figures need to be in grey scale with altered shades or patterns
- tables require correct symbols for prime.
-Raw data was provided
- requires more detail or referencing in methods to enable repetition
The title, hypothesis and model proposed together with the conclusion is overstated.

Experimental design

Original primary research that falls within the scope of the journal.
Research question could be better defined to be relevant & meaningful.
The gap was over generalised thus did not clearly align to the aims and methods.

Methods could be better described (or referenced) to provide sufficient detail & information to replicate. Concentrations for reagents (antibodies, protein loaded, RNA transcribed) are missing.
Rigorous investigation was carried out and performed to a high technical & ethical standard.
Westerns confirmed and complemented gene expression but this was not a clear in the abstract.
Methods described with insufficient detail & information to replicate- doses and rose ranges need clarity, concentrations rather than dilutions, concentrations of protein loaded for westerns needs inclusions.

The cell culture included osteoclasts induced by RANKL and MCF as would be the situation in physiology and pathology.

The link to OP was not clear and it being an OP model was over state. Moving information from the discussion to the intro would have provided a better background an justification for the aims and the study.

Validity of the findings

Impact and novelty was not clear as it related to osteoporosis in general.
The focus would be stronger if was more clearly aligned to the role of the factors investigated in terms of the mechanism of the osteoclast differentiation, not activity.
Studies were replicated for validity and stated.
All underlying data have been provided; they are robust, statistically sound, generally well controlled.
Conclusions were over-stated and do not align to the results presented.


All underlying data have been provided; they are robust, statistically sound, & controlled.
Conclusions were over-stated and could better align to the testable aims and results.

Gene and protein expression in a cell culture situation do not necessarily reflect an osteoclast nor active

I would suggest assessing function or effective resorption by Ca2+ release or bone substrate interaction if considering this as a model of osteoporosis as activity is a feature of menopause induced osteoporosis. If a model of age onset osteoporosis suppression of the osteoblast activity is also a feature.

Additional comments

I append an annotated PDF.

In addition, I have the following observations on your MIQE Checklist which should be resolved:

Line 3: "negative it is not explicit"
Line 9: "not included"
Line 21: "not included:
Line 40: "incorrect prime symbols"

Annotated reviews are not available for download in order to protect the identity of reviewers who chose to remain anonymous.

---

## Round 0.2 · accepted · Accept

Reviewers 1 and 2 have expressed their satisfaction with the revised version. While Reviewer 3 did not provide a final assessment of the revised manuscript, I have thoroughly examined your comprehensive responses to their concerns. You have provided clear explanations for your methodological choices, added necessary details to improve reproducibility, and made significant efforts to enhance the clarity of your manuscript's structure and conclusions. You have incorporated many of Reviewer 3's suggestions, and also provided well-reasoned justifications for maintaining certain aspects of your study design. The additional context and limitations you've agreed to include in the revised manuscript will strengthen the paper's relevance to the field.

Given the thorough nature of your revisions and responses, along with the positive feedback from Reviewers 1 and 2, I am confident in moving forward with the acceptance of your manuscript. Please ensure that all final adjustments, including any formatting requirements, are completed before submission of the final version.

Reviewer 1 ·

Basic reporting

In this study, the authors first induced bone marrow macrophages with chemical reagents to establish an osteoporosis model, and determined and constructed the correlation between Pdk3 and osteoclast formation related genes. After constructing the Pdk3 silencing cell line, the phosphorylation and expression levels of ERK, P65, and JAK/STAT were elucidated through molecular detection, laying the foundation for revealing the molecular mechanisms of osteoporosis mediated by this gene. Subsequently, the expression of apoptosis and inflammation related genes was evaluated by qPCR. In summary, this study aims to investigate the molecular mechanisms affecting osteoporosis, particularly from the perspective of osteoclast phenotype, through a series of cell experiments and molecular detection, in order to reveal the relevant mechanisms. The experimental design is reasonable, the logic is complete, the verification is sufficient, the conclusion is not excessively extended, and the discussion is appropriate. I am very satisfied with the revised manuscript.

Experimental design

no comment

Validity of the findings

no comment

Reviewer 2 ·

Basic reporting

no comment

Experimental design

no comment

Validity of the findings

no comment

Additional comments

In this study, the authors explored the potential link between Pdk3 and osteoclastogenesis in osteoporosis. They found that after exposure to RANKL, the levels of Pdk3 and osteoclast formation characteristic genes increased, and there was a positive correlation between Pdk3 and osteoclast formation characteristic genes. Meanwhile, RANKL increased the phosphorylation of ERK, P65, and JAK/STAT, confirming a positive correlation between Pdk3 and the phosphorylation of ERK, P65, and JAK/STAT. In addition, in RANKL exposed osteoclasts, Pdk3 knockout reduced ERK, P65, and JAK/STAT phosphorylation, decreased expression of osteoclastogenetic characteristic genes, reduced inflammation, and exacerbated cell apoptosis. These results reveal the role of Pdk3 in osteoporosis, laying the foundation for further research on the specificity of Pdk3 for other bone diseases. The overall experimental design is reasonable and rigorous, with new discoveries and a certain degree of innovation, meeting peerj's publishing requirements.